# Validating Atlantic salmon (Salmo Salar) scale reading by genetic parent assignment and PIT-tagging

**Kjell Rong Utne**◉[1]*, **Marine Servane Ono Brieuc**[1], **Per Tommy Fjeldheim**[1], **Kurt Urdal**[2], **Gunnel Marie Østborg**[3], **Kevin A. Glover**[1], **Alison Harvey**[1], **Øystein Skaala**[1]

1 Department of population genetics, Institute of Marine Research, Bergen, Norway, 2 Rådgivende Biolger AS, Bergen, Norway, 3 Department salmonids, Norwegian Institute for Nature Research (NINA), Trondheim, Norway

* kjell.rong.utne@hi.no

## Abstract

Understanding changes in abundance and survival in Atlantic salmon populations requires knowledge of growth rates and age. Salmon are typically aged through scale reading, but such estimates are rarely validated against age-verified fish from the wild. Here, we present a unique dataset of scales from 254 PIT-tagged Atlantic salmon with known sea-age. In addition, the freshwater age is known for 81 of these fish, through genetic parent-offspring identification. This dataset was used to estimate precision and bias in age readings and back-calculated length, as estimated by three independent experienced salmon scale readers. Overall, readers had an accuracy of 97.1% for sea-age and 71.7% for freshwater-age. For sea-age, scale reading was less accurate for salmon that had spent 2 or more years at sea than for salmon that had spent 1 year at sea. Freshwater age did however not affect scale reading accuracy. None of the scale readers erroneously misclassified freshwater- or sea-age with more than one year, and there was no significant pattern of misclassified ages to be under- or overestimate by the scale readers. Back-calculated smolt length was significantly different to length when measured as a smolt prior to seaward migration: it was shorter than the measured body-length for small smolts and longer for large smolts. This unique dataset, including the age-validated images of all scales, is now made openly available providing an important resource for training and testing salmon scale readers globally.

## Introduction

Fisheries management often relies on knowing the age structure of a population and the harvested individuals [1]. Furthermore, studies of life history traits of fish such as growth, age at maturation and size at age, and how these traits vary over time due to changing environmental conditions [e.g., 2,3], all rely on knowing the age of the

**Data availability statement:** All scales images, an excel-sheet with corresponding information, including sea-age and freshwater age (when available), and R-code used calculate the results presented in this study can be downloaded from the following link: https://doi.org/10.6084/m9.figshare.28505054.v1.

**Funding:** The study was funded by the Norwegian ministry for trade, industry and fisheries. The funders had no role in study design, data collection and analysis, decision to publish, or preparation of the manuscript.

**Competing interests:** The authors have declared that no competing interests exist.

fish. The age of a sampled fish can be estimated from scales, otoliths, spines or other hard structures [4]. The structure used by researchers to estimate age of sampled individuals varies among species, but scales or otoliths are often preferred as they are easily retrieved and provide relatively precise age estimates with low bias.

Both otoliths and scales are used to estimate age of Atlantic salmon (*Salmo Salar*) [5], but scale sampling is usually preferred as it is non-invasive and individuals can be released after one or a few scales are retrieved. In addition, for recreational anglers providing data to scientists, it's easier to sample scales than otoliths. Salmon scales have multiple circuli (also called sclerites), which are deposited as the fish grows and form a dark circle in the scale. The distance between circuli is positively associated with body growth rate [6,7]. An area with narrow bands between circuli is defined as a "winter-zone", as the growth is slower during the winter than the summer [8]. Estimating age by scale reading is done by counting the winter-zones along the anterior-posterior axis of a scale [9,10] and has been done for salmon for more than a century [11]. Disaggregated data of sea-age for harvested salmon form the basis for the stock assessment of Atlantic salmon [12]. Furthermore, estimated age can together with other morphological data from salmon provide knowledge about temporal and spatial trends in marine growth [13–15], freshwater growth [16] and life-history strategies [e.g., 2,17].

Age reading of scales has traditionally been performed by trained personnel, even though automatic age estimations performed by computers are also being developed [18,19]. Estimating age from fish scales manually also involves some degree of subjective decisions, with a risk of wrongly estimating the age of the fish. Previous work on salmonids have estimated age estimation accuracy for chum (*Oncorhynchus keta*) and sockeye salmon (*Oncorhynchus nerka*) [5], brown trout (*Salmo trutta*) [20,21] and steelhead (*Oncorhynchus mykiss*) [22], where the age of the fish was known from mark-recapture experiments or from parent-offspring relationships and known spawning times. Although estimates of precision and bias for estimated age of manually read Atlantic salmon scales have been reported [23,24], no studies on this topic have been published in peer-reviewed journals since the work by Havey in 1959 [25]. A reason for the limited number of studies addressing this issue is the lack of sampled scales from wild salmon with known age. Furthermore, validated age-data for the entire range of longevity is lacking not only for salmon, but for fish in general [26].

Here, we present a unique dataset of scale images from 254 wild Atlantic salmon returning to the river Etneelva located on the western Norwegian coast (59˚41'N, 5˚56' E). The salmon were tagged with Passive Integrated Transponder (PIT) tags and a tissue sample was taken when leaving the river as smolts. The river has a resistance board weir fish trap installed in the lower part of the river to monitor and sample returning adult salmon. The number of years spent at sea was determined directly for the recaptured PIT-tagged returning salmon. Furthermore, as tissue samples for DNA profiling are taken from all adult returning salmon before they could proceed to the spawning grounds [27,28], one or both parents for some of the smolts were identified using genetic parentage assignment. Since the year of spawning is known for the parents, the number of years the smolts had spent in the river could be

 

calculated for 81 of the 254 individuals. Hence, this unique dataset contains individuals with both known freshwater-age and sea-age and is herein described in detail and made available as open-access. The objective of the study was to quantify the precision and bias of three experienced scale readers who estimated freshwater and sea-age from the wild Atlantic salmon scales presented in this study. The hypothesis was that scale reading accuracy does not vary with the salmon's freshwater- or sea-age. A second objective was to estimate whether back-calculated smolt length deviate from measured smolt length. The hypothesis was that back-calculated length is not significantly different from measured smolt length.

## Materials and methods

### River sampling

**Adult salmon.** The scales were retrieved from adult salmon returning to the river Etneelva in western Norway (59° 41'N, 5°56' E). A resistance board weir fish trap in the river leads the salmon into a cage, and both scale and tissue (fin clip) samples are taken from the salmon before they are released above the trap and can continue to the spawning grounds (Harvey et al. 2017). The present study is based on samples from 254 adult salmon sampled in Jun-Aug 2017–2022 that had been PIT tagged as smolts (see below). The length (nearest cm) and weight (nearest 10 gram) were measured from each salmon at the time of sampling, a micro-clip was taken from the tip of the adipose fin for genetics analyses, and multiple scales were sampled 3–6 rows above the lateral line, and on a line extending from the anterior edge of the anal fin to the posterior edge of the dorsal fin for later age-reading. The sampled fish were within the range 45.5–108 cm, 0.73–12.03 kg and had spent 1–5 years in the sea before returning to the river (Table 1b and S1 Fig). None of the sampled salmon were anesthetized or killed, and all tissue sampling and size measurements were carried out swiftly in order to minimize handling stress during this process. All fish sampling and handling as carried out according to the recommendations of the Norwegian Animal Research Authority (NARA), and the sampling was approved by NARA (approval/permit number 30061).

**Smolts.** The river also has a smolt trap positioned in the lower part of the river, which intercepts a proportion of the smolts migrating from the river each spring and early summer. Some of those smolts are PIT tagged and their body length (nearest 0.1 cm) is measured before they are released in the river to continue their migration into the sea. Of the total dataset of 254 PIT-tagged returning adult salmon, the smolt length had been measured for 248 individuals. The number of years spent at sea (sea-age) could be calculated by subtracting the smolt year from the year the salmon returned to the river (Table 1).

**Identifying parent-offspring relationships.** Unlike sea-age, freshwater-age cannot be determined by direct comparison through tagging. Most adult salmon entering the Etneelva river have been sampled for genetic analysis since 2013 and genotyped at 31 microsatellite markers as described in Harvey et al. [28]. This presents a rare opportunity to reliably estimate freshwater-age for a subset of individuals through pedigree reconstruction, where the parents of an

**Table 1. A) Smolt length (248 individuals) and freshwater-age (81 individuals), B) Body length, weight, and sea-age for returning salmon (254 individuals).** Provided in the table are measured average, minimum, maximum, 5% and 95% quantiles.

|  | Average | 5% | 95% | Min | Max |
|---|---|---|---|---|---|
| A) |  |  |  |  |  |
| Length (cm) | 13.73 | 12.00 | 15.56 | 11.50 | 17.00 |
| Age (years) | 2.54 | 2 | 3 | 1 | 5 |
| B) |  |  |  |  |  |
| Length (cm) | 69.12 | 52.30 | 91.80 | 45.50 | 108 |
| Weight (kg) | 3.38 | 1.20 | 7.34 | 0.73 | 12.03 |
| Age (years) | 1.61 | 1 | 3 | 1 | 5 |

individual fish can be identified when the offspring and both parents are sampled for genetics. We attempted to identify the parents of the 254 returning adults mentioned above using Colony v.2.0.7.1 [29] and all adults from the relevant years as candidate parents. To ensure reliable estimates of freshwater-age, we only retained individuals where both parents were identified through pedigree analysis, and only if those had returned in the same year. The offspring hatch the year after mating of the parents, therefore freshwater-age was calculated as followed: freshwater-age = smolt-year – return-year of the parents – 1. Parentage assignment reliably identified 81 parent-offspring trios. The freshwater age of these 81 individuals varied between 1 and 5 years, but most of the individuals had spent 2 (n = 38) or 3 years (n = 40) in the river.

**Scale sampling and age reading.** Up to five scales were retrieved from each adult salmon, but only one scale per individual was included in the final dataset. This was based on a prior selection considering the general quality of the scale, and how easy individual circuli could be distinguished. The scale image was read by three independent readers, each with multiple years of experience reading salmon scales. The readers each have 8–26 years of experience reading salmon scales. The three readers work for different research organizations located in three different cities, and do not collaborate on a regular basis. The readers have however collaborated in earlier projects and have therefore been able to synchronize their methods and age reading evaluations in previous years. Scale reading was carried out according to international guidelines for scale reading in Atlantic salmon [30,31]. All three scale-readers estimated sea-age and smolt length. In addition, two of the scale-readers estimated freshwater-age as well as the winter body-length for both the freshwater- and sea-phase.

Salmon scales were dried, put on a glass plate and photographed with a compound microscope (Nikon SMZ 1500) fitted with a digital camera (Nikon DS-FI3). The age (both sea- and freshwater-age) was set by counting the number of dark annuli on the scale (for scale reading protocols see [30]). The body-length at a given age was back-calculated from measurements of the distance (in mm) from the center of the scale to the circuli representing the given age (Fig 1). The selection of the circuli representing each age is a somewhat subjective decision made during scale reading based on when circuli spacing increased, representing the end of the freshwater phase or the end of the winter phase. Smolt length

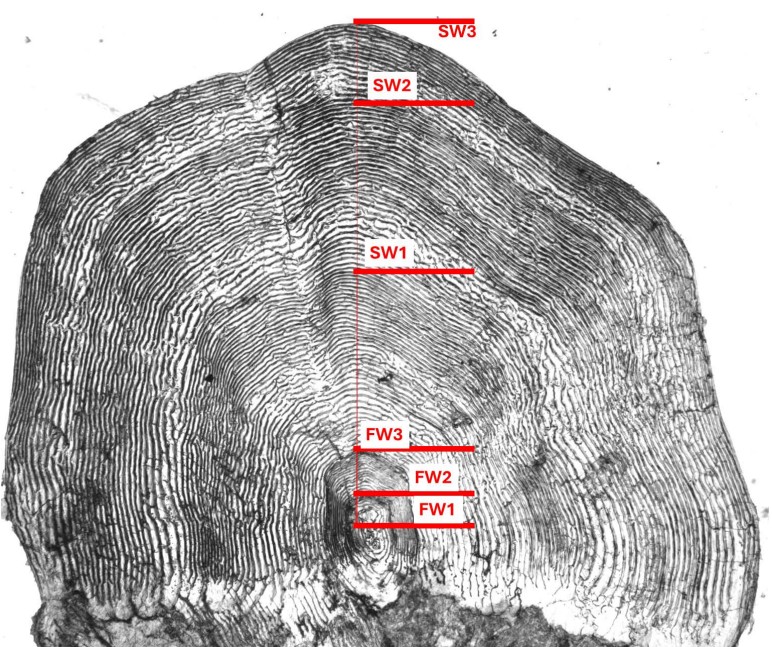

**Fig 1. Annotated scale image.** An example of a salmon scale with annotated freshwater- and sea-age set by one of the expert scale readers. As evident in the picture, the freshwater-age was 3 years, and the sea-age was 3 years.

was defined as the last circuli deposited in freshwater and for length-at-age as the last narrow circuli defining the end of the winter period for each year.

Body-length was back-calculated using the Lea-Dahl method, which assumes a linear relationship between scale radius and body length throughout the life-span of an individual fish [30]. Body length at a given time is calculated from scale measurements with the following equation

$$L_t = \frac{S_t}{S_c}(L_c)$$

where $L_t$ is the body length at a time $t$, $L_c$ is the body length at capture, $S_t$ is the radius of the scale annulus at time $t$ and $S_c$ is the total scale radius. Winter body-length was calculated from the last scale annulus deposited in the dark winter band (for more information see [30]). In addition to scale images, the reader had access to information of body length, weight, sex and catch date for the sampled salmon, which could guide their evaluation. The measured growth in body-length defined as the difference between smolt length and length-at-return, and the back-calculated growth defined as the difference between back-calculated winter body-length and smolt length, are given in S1 Table.

## Statistical analyses

Scale reading precision has traditionally been estimated with the coefficient of variation (CV) or alternatively with the average percentage error (APE) [32]. Both these indices are a measurement of error relative to the true age of the fish, where an error of one year has a relatively larger effect on the indices for young fish than for old fish. The use of CV or APE to measured precision is reasonable for a wide age distribution with a significant proportion of relatively old fish. However, 99% of the scales in the dataset of the present study came from salmon with a true sea-age of 1–3 years. We define all misclassified freshwater- and sea-ages as equally important irrespective of their true age and estimate the proportion of incorrectly classified ages (PE) with the following equation

$$PE = \frac{100}{n} \sum_{i=1}^{n} \frac{|x_i - y_i|}{x_i}$$

where $n$ is the number of scales read and estimate age for scale number $i$ is 0 when identical to true age and 1 otherwise. A 95% confidence interval for the estimated PE was calculated by a bootstrapping approach, by sampling with replacement 1000 times from the original dataset and retrieving the 2.5% and 97.5% quantiles of the 1000 estimated PE's. The calculations were done separately for sea- and freshwater-age.

The precision of the length measurements was estimated with the coefficient of variation (CV) which is unitless and calculated with the following equation [32]

$$CV = \frac{100}{n} \sum_{j=1}^{n} \frac{\sqrt{\sum_{i=1}^{r} \frac{(x_{ij} - x_i)^2}{r-1}}}{x_j}$$

where $n$ is the number of scales read, $r$ is the number of age readers, $X_{ij}$ is the length measured by reader number $i$ for scale number $j$. $X_j$ is the true length of the fish measured as smolts and the mean of the scale-reader measurements when the true length is unknown. A 95% confidence interval for the estimated CV was calculated by bootstrapping, by sampling with replacement 1000 times from the original dataset and retrieving the 2.5% and 97.5% quantiles of the 1000 estimated CV's. The CV was calculated for freshwater-age 1–3 and sea-age 1–3. For higher freshwater- and sea- ages, there were an insufficient number of estimates to calculate CV.

Evans and Hoenig test of asymmetry was applied to estimate any disagreements between reader estimates and true freshwater- or sea-age [33]. This test estimates whether scale readers either under- or overestimate the true age, where a p-value < 0.05 indicates asymmetric errors.

A Chi-square test was used to test if back-calculated smolt length was significantly over- or underestimated compared to the measured smolt length. The Chi-square test was applied for the back-calculated smolt length read by at least one scale reader, with the average back-calculated length applied for scales read by two or three readers. A Chi-square test was also applied for the back-calculated smolt length for each individual reader. A linear regression was used to estimate the relationship between back-calculated and measured smolt length, and a two-sided t-test with α-level of 0.05 was used to test if the regression slope deviated from 1.

The estimated fresh- or sea-age set by the scale readers was as either correct or incorrect, and therefore follow a binomial distribution. A generalized logistic regression was applied to test if the probability to correctly estimate the sea-age ($Sea\_Age_{acurray}$) based on scale readings was correlated to salmon size, sex or growth. The model had the following form

$$Sea\_Age_{accuracy} = sea\_age_{corr} * bl + year_{smolt} + bl_{smolt} + bl_{adult} + sex + \varepsilon$$

Where $sea\_age_{corr}$ is the true sea-age aggregated into 1SW (1 sea winter – 1 year at sea) or MSW (Multi-sea winter – 2 or more years at sea), $year_{smolt}$ is a factor for the year the smolt entered the sea, $bl_{smolt}$ is the body-length as smolt, $bl_{adult}$ is the body-length when returning to the river and $\varepsilon$ is a normal distributed error term. $Sea\_Age_{accuracy}$ is 0 when all readers correctly estimated the sea-age and 1 if at least one reader had estimated the wrong sea-age. The aggregation of salmon into the MSW-group, representing salmon with sea-age of 2–5 years, was chosen due to the low number of scales from salmon that had spent more than 2 years at sea (n = 23). A generalized logistic regression was also applied to test if the probability to correctly estimate freshwater-age ($FW\_Age_{acurray}$) based on scale readings was correlated to smolt sex, size or freshwater age. The model had the following form

$$FW\_Age_{accuracy} = fw\_age_{corr} + year_{smolt} + bl_{smolt} + sex + \varepsilon$$

Where $fw\_age_{corr}$ is the true freshwater-age grouped into 1–2 and 3–5 years. The parsimony principle was used to select the best model from all possible combinations of explanatory variables by selecting the model with the lowest Akaike Information Criterion (AIC). Model performance was evaluated by fitting model residuals against covariates.

All statistical analyses were performed in the software *R* version 4.3.2 [34] applying the following packages; lme4 [35], ggplot2 [36], FSD [37] and DHARMa [38].

## Results

### Sea age

The percent agreement between known sea-age and estimated sea-age from scale-readings was 97.1% (95% C.I: 95.7–98.2%). The readers had a similar accuracy as each of the three scale readers misclassified the sea-age of just 6–9 of the 254 salmon. None of the readers erroneously misclassified the sea-age by more than one year, and there was no indication of systematic bias in the age-readings, measured as a tendency to under- or overestimate the sea-age when combining the estimates from all three readers (Evans Hoenig test, $\chi 2 = 1.43, df = 2, p = 0.49$, S2 Fig), nor for any of the three individual readers (Evans Hoenig test, $p > 0.05$).

The probability that at least one scale-reader misclassified the sea-age was higher for salmon that had spent more than one year at sea (glm, z-value = 1.722, p = 0.027). The percentage misclassified sea-age was 1.7% (2/126) for salmon with sea-age 1 and 8.7% (11/128) for salmon with sea-age 2 or 3. The other covariates included in the original model; sex,

body-length as smolt, body-length as returning adults and year of sea-entry did not affect the probability of misclassification of sea-age by the scale-readers.

### Freshwater age and back-calculated smolt-length

Two of the three scale readers estimated winter body-length for the years the salmon stayed in the river as parr, as well as for the marine phase. The true freshwater-age was known for 81 individuals based on genetic identification of parent-offspring relationships. Both readers abstained from estimating the freshwater-age for ~1/3 of the scales taken from salmon with known freshwater-age, as the readers could not identify winter-rings with sufficient certainty and therefore deemed the scales as "unreadable".

The percentage agreement between known freshwater-age and estimated age from scale-readings was 71.7% (C.I: 63.3%-80.2%). The two scale-readers set the correct freshwater-age for 76.9% (40/52) and 66.7% (36/54) of the scales. None of the readers erroneously misclassified the freshwater-age by more than one year, and there was no indication of bias in the age-readings, measured as a tendency to under- or overestimate the freshwater-age (Evans Hoenig test, $\chi2 = 0.13$, df = 1, p = 0.72, S2 Fig). The probability for at least one scale-readers to misclassify the freshwater-age was not affected by the salmon's sex, freshwater-age, smolt length or smolt year (glm, z-value<1, p>0.05).

All three scale-readers estimated the back-calculated smolt length, but also abstained from providing an estimate for a subset of the scales for which they did not feel sufficiently confident defining the smolt length. However, the number of scales deemed "unreadable" for smolt length varied greatly among the scale readers, as the three scale-readers abstained from estimating smolt length for 8, 61 and 111 of the 248 scales with known smolt length. For the scales with estimated smolt length by at least one scale-reader, the average back-calculated smolt length was 0.27 cm shorter than the measured smolt length, which was significantly different from the measured smolt length (t.test = -2.6581, df = 240, p = 0.008). The associated CV for smolt length was 9.36 (95% C.I: 8.27–10.61). The probability for the back-calculated length to be shorter (134 of 241) or longer (107 of 241) than the measured smolt length was not significantly different (chi-square test statistic $\chi^2 = 3.02$, df = 1, p = 0.08) when aggregating the results from all three scale-readers. However, two of the scale-readers significantly underestimated the back-calculated length while one scale-reader significantly overestimated the back-calculated length (see supplement, S3 Fig and S2 Table). Furthermore, the estimated slope of the linear regression between back-calculated length and measured length was 1.26 with a standard error of 0.09, which is significantly different to a slope of 1 (regression t-test(239)=14.575, p<0.001). Hence, for the smallest smolts the back-calculated length was on average shorter than the measured length, while for the largest smolts the back-calculated length was on average longer than the measured length (Fig 2). The CV of the estimated winter body-length was from 0.62 (CI: 0.36–0.94) to 9.07 (CI: 7.73–10.66) and decreased with freshwater- and sea-age (Table 2).

## Discussion

Understanding changes in abundance and survival in fish populations requires knowledge of growth rates and individual age. However, the accuracy of scale reading to provide these data has rarely been tested. Here, we present a unique dataset of Atlantic salmon scales with known river- and sea-age, with annotated images of the salmon scales made available as open-access in the supplementary information. The dataset is therefore available for anyone interested in training or testing their salmon scale readings skills against data with known age.

The scales from salmon with verified age were used to estimate precision and bias in age and back-calculated length for Atlantic salmon estimated by three independent experienced salmon scale readers. Overall, the accuracy in estimated sea-age was high (97.1%), which is within the range of estimates of sea-age for other salmonids, where the accuracy was 93–96.9% for steelhead, chum and sockeye salmon [5,22]. In the study by Havey [25], only ~80% of the sampled Atlantic salmon had the correct estimated sea-age, but most of the age-estimation errors were extra annuli associated

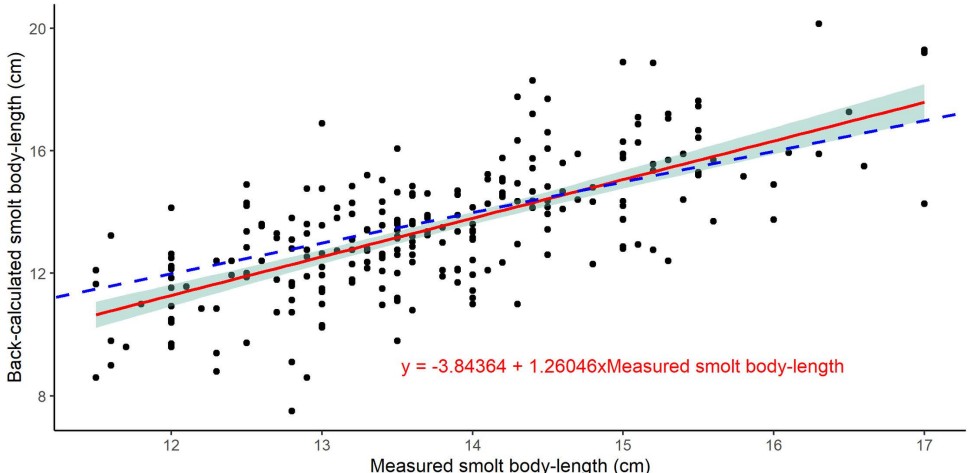

**Fig 2. Measured and back-calculated smolt length.** Relationship between smolt body-length measured on fish migrating from the river, and estimated from back-calculated scale readings. The blue dotted line shows the 1:1 relationship between measured and back-calculated smolt length. The red line shows the linear regression between back-calculated length and measured length, together with 95% confidence intervals for the regression (shaded area).

**Table 2. Estimated coefficient of variation (CV) with 95% confidence intervals (C.I) for freshwater- and sea age.**

| Category | CV | C.I 2.5% | C.I 97.5% |
|---|---|---|---|
| Freshwater-age 1 | 9.07 | 7.73 | 10.66 |
| Freshwater-age 2 | 5.87 | 4.53 | 7.38 |
| Freshwater-age 3 | 2.29 | 1.62 | 3.24 |
| Sea-age 1 | 1.70 | 1.27 | 2.30 |
| Sea-age 2 | 1.29 | 0.90 | 1.72 |
| Sea-age 3 | 0.62 | 0.36 | 0.94 |

with hatchery-rearing conditions, and therefore not associated with annuli deposited after the first winter at sea. The high accuracy estimated in the present study could be related to the life-history of Atlantic salmon with rapid marine growth and few years spent at sea, making it easier to estimate sea age than for species with slow growth and higher longevity. The accuracy reported from this study is higher than previous estimates for Atlantic salmon presented in the gray literature, which reported an individual scale reader accuracy in the range 43–92% [39] and a reader agreement of 92% for salmon of unknown age [24]. The present study was based on the image of only one scale from each salmon, and the scale reader accuracy could potentially be higher if the scale readers would base their estimates on multiple scales. The study nevertheless fills a significant knowledge gap since age estimation accuracy for Atlantic salmon has not been addressed in a peer-reviewed paper since the study by Havey in 1959 [25].

Scale reader accuracy was lower for freshwater-age (71.7%) than for sea-age (97.1%). Slower growth in freshwater makes it more challenging to correctly determine the age of salmon for the river phase than for the marine phase. However, the accuracy for freshwater-age in this study was lower than for instance earlier age-readings of brown trout in freshwater, where the reported accuracy was 93.6% [20] and 79.2% [21]. The scale reading accuracy for freshwater-age seems to increase with body length and/or growth rates. The present study included salmon smolts with a body-length of 11.5–17.0 cm and a scale reader accuracy of 71.7%. The study by Vaorka et al. [21] of brown trout with a body length of 7.3–27.6 cm had a scale reader accuracy of 79.2% while Rifflart et al. [20] reported an accuracy of 93.6% for brown trout

when the study mainly included individuals with a body length of 20–30 cm. The focus of the presented study included both the freshwater- and the marine growth, and the images of the entire scales might result in too low resolution for the readers to clearly separate circuli deposited in freshwater. Hence the accuracy of the estimated freshwater-age will probably be higher if the scale images only include the center of the scale, thereby increasing the pixel resolution for the part of the scale representing the freshwater phase.

The three scale readers all had multiple years of experience. Furthermore, they all routinely work with age-reading salmon from western Norway, the area from which the scales in this study originated. Less experienced scale readers should be expected to have a lower age-reading precision. Furthermore, precision is most likely affected by how familiar the age-reader is with the growth pattern and age distribution of the sampled salmon, which partly varies for salmon originating from different parts of the Atlantic Ocean [12]. Scale readers will have to decide whether the age can be estimated from a given scale and this decision is partly subjective. For a replacement scale, a scale which has replaced a scale lost during the life-cycle, the age cannot be estimated due to lack of circuli for years before the replacement. In other cases, the scale may have been damaged, for instance during sampling. The "unreadable" scales are typically assumed to originate randomly from the total sample of fish. If scales with unclear circuli are discarded due to difficulties in estimating the age, this can increase the possibility to discard scales from old or slow-growing individuals as these are often the most difficult to read, which would lead to a bias in the estimate age distribution or average growth for the population. The readers in this study were instructed to read all the scales, but some scales defined as unreadable were removed from the initial dataset after a discussion with all readers. However, the three scale readers had very different criteria for whether the freshwater-age and back-calculated smolt length could be estimated from a given scale. The definition of unreadable scales should be stated both during routine readings and when evaluating the accuracy of age readings.

There was a higher probability to estimate the wrong sea-age for MSW salmon than for 1SW salmon. The majority of the surviving Atlantic salmon return to their home-river after one year at sea [12], and the management impact of wrongly estimated sea-ages between 2- and 3-year-old salmon is likely to be limited. For studies addressing for instance variation in life-history strategies (e.g., review by [40]), the effect of wrongly estimated sea-ages can have a more severe impact on the result. The accuracy for estimated freshwater-age was lower than for the estimated sea-age. Furthermore, the scale-readers defined more of the salmon scales as "unreadable" for freshwater-age than for sea-age, demonstrating the higher uncertainty for freshwater-age compared to sea-age. Although there was no systematic bias for the estimated freshwater-age, the results demonstrate the associated uncertainty with these measurements. Changing environmental conditions affecting salmon growth patterns can affect the accuracy and precision of scale age estimates. Scale circuli deposition vary with water temperature [6], and parr in the river Tana/Teno has over several decades experienced higher water temperatures and slower freshwater growth [16]. In such cases, where growth rate decreases, smolt age increases, and circuli deposition rate potentially increases, it will most likely be more difficult to estimate the correct freshwater-age from the scales. For the marine phase, the presented results suggests that the proportion of misclassified salmon sea-ages varies with sea-age, and a higher proportion of salmon with delayed maturation [14,17] will result in an overall slightly lower accuracy for the estimated sea-age.

Several studies have investigated temporal changes in marine growth using back-calculated length estimated from scales of salmon returning to rivers [e.g., 3, 41, 42]. Such measurements have also been instrumental in looking at plasticity of the *Vgll3* and *Six6* genes and the interaction between individual growth rates and degree of effect of these genes on age at maturation [17]. Although it is not known if estimated growth presented in these studies were biased, there is a potential for bias with back-calculated growth, as previously demonstrated by Hanson et al. [43] who proposed a correction factor for the back-calculated length using the Lea-Dahl method. Furthermore, the bias in back-calculated length is not only affected by the applied method, but also by genetic predisposition for age at maturation [44]. Hence, salmon genetically predisposed to delay maturation have smaller scales for their body

 

size than salmon genetically predisposed to mature early. We found no systematic bias in whether the scale readers over- or underestimated the sea-age nor the freshwater-age. However, the present study demonstrates that the back-calculated smolt length deviated from the measured smolt length. While the body-length of the smolt with the shortest measured body-length was underestimated, the body-length of the largest smolts was overestimated. The results therefore show a similar, although smaller, bias for back-calculated length with the Lea-Dahl method as previously documented by Hanson et al. [43]. The correction equation proposed by Hanson et al. [43] overcompensates for the bias estimated in this study (calculations not presented here), showing that a general correction factor must only be used with caution.

The presented study was based on 254 salmon scales with known sea-age, and 81 scales with known freshwater-age. Although there are other studies of scale reading accuracy for Atlantic salmon with sample sizes below 200 scales [25,39], several studies addressing accuracy and precision in age estimation of anadrome fish have been based on between 1000 and 10 000 scales [e.g., 5,18,21,22]. The general conclusions of the study presented here are not hindered by a rather low sample size, although a larger dataset could provide more information about how age estimation accuracy varies with fish size, growth patterns or the environmental conditions experienced by the salmon. For instance, the results presented here showed a higher probability of an incorrectly estimated sea-age for MSW salmon than for 1SW salmon. Few scales of salmon with sea-age above two years in the available dataset restricted additional detailed analyses of scale reading accuracy for MSW salmon.

## Supporting information

**S1 Fig. Sampled individuals.** Biological information for the sampled salmon included in this study. A) Measured smolt length, B) True freshwater-age, C) Body-length when returning to the river, D) Body weight when returning to the river, E) True sea-age.
(TIFF)

**S1 Table. Back-calculated length growth.** Minimum, maximum and average measured and back-calculated length growth. Measured growth is calculated as the difference in body-length between the length at return and smolt length, while back-calculated growth is the difference in back-calculated length at the end of a winter sone and smolt length. Year 1 represents the growth until the end of the first winter-zone, year 2 represents the plus growth gained during the spring before entering the river (for 1SW salmon) or until the end of the second winter zone (for MSW salmon), year 3 represents the plus growth gained during the spring before entering the river (for 2SW salmon) or to the end of third winter zone (for 3SW salmon). None of the 3SW salmon had plus growth the last spring at sea.
(DOCX)

**S2 Table. Evans Hoenig test table.** The number of smolt lengths under- and overestimated by each of the three scale-readers, as well as $\chi^2$ and p-values estimated with the Evans Hoenig test.
(DOCX)

**S2 Fig. Evans Hoenig test figure.** The difference between the true freshwater-age (a) and seawater-age (b) against the corresponding estimates by the scale readers. The circles represent average difference while bars represent the corresponding 95% confidence intervals.
(TIF)

**S3 Fig. Smolt length and back-calculated length.** The relationship between measured smolt length (cm) and back-calculated smolt length (cm) by each of the three scale-readers.
(TIFF)

## Acknowledgments

This study benefited from the work of multiple colleagues involved in river monitoring, data sampling and fish tagging. We would like to acknowledge Vidar Wennevik for comments on this work, and vital input on the approach to validate freshwater age.

## Author contributions

**Conceptualization:** Kjell Rong Utne, Per Tommy Fjeldheim, Kevin A. Glover, Alison Harvey, Øystein Skaala.

**Data curation:** Marine Servane Ono Brieuc, Per Tommy Fjeldheim, Kurt Urdal, Gunnel Marie Østborg.

**Formal analysis:** Kjell Rong Utne, Marine Servane Ono Brieuc, Per Tommy Fjeldheim, Kurt Urdal, Gunnel Marie Østborg.

**Visualization:** Kjell Rong Utne.

**Writing – original draft:** Kjell Rong Utne, Marine Servane Ono Brieuc, Per Tommy Fjeldheim, Kurt Urdal, Gunnel Marie Østborg, Kevin A. Glover, Alison Harvey, Øystein Skaala.

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
