## [Decision Letter · Decision Letter 0]

16 Jan 2025

PONE-D-24-56358Validating Atlantic salmon (Salmo Salar) scale reading by genetic parent assignment and PIT-taggingPLOS ONE

Dear Dr. Utne,

Thank you for submitting your manuscript to PLOS ONE. After careful consideration, we feel that it has merit but does not fully meet PLOS ONE’s publication criteria as it currently stands. Therefore, we invite you to submit a revised version of the manuscript that addresses the points raised during the review process.

1. This manuscript not technically sound, and the data cannot support the conclusions. PLOS ONE is designed to communicate primary scientific research, and welcome submissions in any applied discipline that will contribute to the base of scientific knowledge. But this manuscript not adhere to the criteria for scientific research article that results show not sufficient to support the conclusion.

2. This manuscript has the statistical analysis problem.

3. This manuscript needs to adhere the PLOS Data Policy. The authors need to make all methods, materials and data underlying the findings in their manuscript fully available.

4. The revised manuscript needs to address each of the comments of the reviewers.

We look forward to receiving your revised manuscript.

Kind regards,

Tzong-Yueh Chen, Ph.D.

Academic Editor

PLOS ONE

Journal Requirements:

“The study was funded by the Norwegian ministry for trade, industry and fisheries. “

4. Thank you for uploading your study's underlying data set. Unfortunately, the repository you have noted in your Data Availability statement does not qualify as an acceptable data repository according to PLOS's standards.

5. We notice that your supplementary [Table S1] are included in the manuscript file. Please remove them and upload them with the file type 'Supporting Information'. Please ensure that each Supporting Information file has a legend listed in the manuscript after the references list.

Reviewers' comments:

Reviewer's Responses to Questions

**Comments to the Author**

1. Is the manuscript technically sound, and do the data support the conclusions?

Reviewer #1: Yes

Reviewer #2: Partly

2. Has the statistical analysis been performed appropriately and rigorously? 

Reviewer #1: Yes

Reviewer #2: N/A

3. Have the authors made all data underlying the findings in their manuscript fully available?

Reviewer #1: Yes

Reviewer #2: No

4. Is the manuscript presented in an intelligible fashion and written in standard English?

Reviewer #1: Yes

Reviewer #2: Yes

5. Review Comments to the Author

Reviewer #1: This manuscript, titled “Validating Atlantic Salmon (Salmo salar) Scale Reading by Genetic Parent Assignment and PIT-Tagging,” effectively evaluates and presents an efficient and accurate method for determining the age of Atlantic salmon through scale reading. Additionally, the authors employed genetic parentage assignment and PIT-tagging to provide a more precise evaluation of freshwater and sea-age distinctions. The manuscript is well-written in fluent English, and the results are both compelling and deserving of publication in PLOS ONE. However, there are a few minor comments for the authors, as outlined below.

1. Why was only one scale included in the final dataset (L131–L132)? Would it not be better to conduct replicates or triplicates to reduce measurement error?

2. This is just a personal comment: Both scales and otoliths are commonly used to determine the age of teleosts. Considering the challenges in distinguishing freshwater and sea age, as mentioned by the authors, why not attempt a comparison with otolith data? Otoliths can effectively differentiate habitats through isotope ratios, such as the Sr:Ca ratio.

Reviewer #2: Manuscript: PONE-D-24-56358

Title: Validating Atlantic Salmon (Salmo Salar) Scale Reading by Genetic Parent Assignment and PIT-Tagging

General Comments:

This paper provides an important dataset and addresses a critical question in fisheries science. However, to enhance its scientific rigor and broader applicability, the authors should improve the description of the experimental design, conduct a more in-depth and complete data analysis, and critically evaluate the implications of their findings. By addressing these issues, this manuscript has the potential to be a valuable contribution to the field.

Specific Comments:

1. The manuscript does not clearly articulate a hypothesis or research question. It is recommended that the authors explicitly state the hypothesis or objectives to provide a clearer research direction.

2. Provide detailed information on whether the sample size of 254 fish is sufficient to generalize the findings, either through statistical power analysis or reference to previous literature.

3. Include more detailed information on the distribution of fish characteristics, such as the precise sites of scale sampling, freshwater age, marine age, and size. A comprehensive description of these properties should be included in the Materials and Methods section.

4. Add images or diagrams to enhance reader understanding and provide clearer visualizations to highlight key findings and address potential biases.

5. Strengthen the manuscript by incorporating additional data, such as more detailed growth and environmental information for fish with known ages. Consider employing more robust validation methods, such as otolith analysis or independent genetic markers.

6. Provide details on controls or calibration methods used in the scale-reading process to minimize or control variability among readers.

7. Since the study relies heavily on experienced readers for scale analysis, include information on how their training was standardized. Discuss calibration procedures to ensure consistency and reliability among readers.

8. Elaborate on the steps taken to verify the accuracy of salmon back-calculated lengths using alternative methods or indicators.

9. Some claims made in the manuscript lack sufficient data support. For instance, the accuracy of Multi-Sea Winter (MSW) salmon is only briefly discussed. Expand on these findings with additional data or analysis.

10. Provide more detail about the statistical methods used. For example: Explain why generalized logistic regression was chosen and how it was validated. Justify the use of Pearson correlation analysis and discuss its limitations in this context.

11. The Discussion section mainly restates results without critically analyzing their implications. Include discussions on potential biases and limitations, such as reader variability or environmental factors affecting scale formation.

12. Situate the study within the broader context of fisheries science. Compare the findings with previous research on Atlantic salmon or other salmonids, highlighting similarities and differences.

6. PLOS authors have the option to publish the peer review history of their article (what does this mean? ). If published, this will include your full peer review and any attached files.

**Do you want your identity to be public for this peer review?** For information about this choice, including consent withdrawal, please see our Privacy Policy .

Reviewer #1: No

Reviewer #2: No

---

## [Author Response · Author response to Decision Letter 0]

11 Mar 2025

Response to reviewers - manuscript (PONE-D-24-56358) “Validating Atlantic salmon (Salmo Salar) scale reading by genetic parent assignment and PIT-tagging”

Reply: We would like to thank the reviewers for the time and effort spent on evaluating the manuscript and for their suggestions for improvements. In general, we agree with the reviewers and have modified the manuscript accordingly. There are no suggestions that we disagree upon, but there are comments that we cannot address due to lack of data (such as including otoliths or multiple scales) or comments we found unclear (such as the request for additional analyses for back-calculated length). Below is a response to each specific comment.

Reviewer 1:

1. Why was only one scale included in the final dataset (L131–L132)? Would it not be better to conduct replicates or triplicates to reduce measurement error?

Reply: Estimating the age of multiple scales from the same fish could improve precision, depending on the general quality of the scales. The study was based on a pre-selected scale with the considered best quality among the sampled scales from each individual fish. The study will not be redone with multiple scales, but a sentence regarding this issue is added to the discussion L330-334.

2. This is just a personal comment: Both scales and otoliths are commonly used to determine the age of teleosts. Considering the challenges in distinguishing freshwater and sea age, as mentioned by the authors, why not attempt a comparison with otolith data? Otoliths can effectively differentiate habitats through isotope ratios, such as the Sr:Ca ratio.

Reply: Comment noted and is relevant for later studies. One issue with otolith data for the river Etneelva is that all salmon should be released alive after sampling. Hence, otoliths cannot be retrieved. It could be possible to retrieve otoliths for individuals caught in the recreational fishery in the river, for the years when it’s opened up for catch-and-kill (most years until now), but otoliths haven’t been retrieved until now. For other rivers in the region, it is not possible with parent-offspring pedigree analysis.

Review 2:

General Comments:

This paper provides an important dataset and addresses a critical question in fisheries science. However, to enhance its scientific rigor and broader applicability, the authors should improve the description of the experimental design, conduct a more in-depth and complete data analysis, and critically evaluate the implications of their findings. By addressing these issues, this manuscript has the potential to be a valuable contribution to the field.

Specific Comments:

1. The manuscript does not clearly articulate a hypothesis or research question. It is recommended that the authors explicitly state the hypothesis or objectives to provide a clearer research direction.

Reply: The last part of the introduction, which addresses objectives and hypothesis, is partly modified. See changes in L82-88.

2. Provide detailed information on whether the sample size of 254 fish is sufficient to generalize the findings, either through statistical power analysis or reference to previous literature.

Reply:

The issue with a rather low sample size is now addressed in the discussion L408-418.

3. Include more detailed information on the distribution of fish characteristics, such as the precise sites of scale sampling, freshwater age, marine age, and size. A comprehensive description of these properties should be included in the Materials and Methods section.

Reply: The text has been updated in L100-104. Furthermore, a table (Tab. 1) and a supplemental figure (S1 Fig) providing information about the size and age of the salmon have been added.

4. Add images or diagrams to enhance reader understanding and provide clearer visualizations to highlight key findings and address potential biases.

Reply: A figure of a salmon scale with marks for freshwater- and sea-age has been added (Fig 1). A figure showing the result of the Evan-Hoenig test for freshwater- and sea-age has been added (S2 Fig).

5. Strengthen the manuscript by incorporating additional data, such as more detailed growth and environmental information for fish with known ages. Consider employing more robust validation methods, such as otolith analysis or independent genetic markers.

Reply:

Information about back-calculated growth is now added as a supplement table referred to in the text. See L172-175, and Table S1.

Second issue – otolith analysis or independent genetic markers. Although we acknowledge the point raised by the reviewer (as well as the first reviewer), and agree that this would strengthen the manuscript, we don’t have the possibility to extend the manuscript with otolith analyses. Otoliths are not routinely sampled from this river, as the fish are released alive after sampling. Independent genetic markers are used in the analyses. We have specified that in L129-130.

6. Provide details on controls or calibration methods used in the scale-reading process to minimize or control variability among readers.

Reply: Sentenced addressing this issue is added to L145-148.

7. Since the study relies heavily on experienced readers for scale analysis, include information on how their training was standardized. Discuss calibration procedures to ensure consistency and reliability among readers.

Reply: Sentenced addressing this issue is added to L145-150.

8. Elaborate on the steps taken to verify the accuracy of salmon back-calculated lengths using alternative methods or indicators.

Reply: The referee comment could be interpreted in two ways.

Firstly, if the referee asks for additional analyses to verify the present back-calculated length, which is obtained using the Dahl-Lea method, we are willing to address that issue. It would then be a great help for us if the referee could suggest alternative methods besides the applied linear model of measured and back-calculated smolt length, the test of whether the slope of the model deviates from 1, and the Evans Hoenig test.

Secondly, if the referee asks for an evaluation of alternative methods to back-calculate length, such as the original or modified Fraiser-Lee method, we are probably able to address this issue. It would however require some additional effort to obtain the data as the scale readers did not report scale radius, but back-calculated length. Testing different back-calculation methods was not an objective of the original study. However, we agree that such analyses would add extra relevance to the manuscript. If the reviewer and editor recommend such analyses to be carried out, we are willing to see if this can be done.

9. Some claims made in the manuscript lack sufficient data support. For instance, the accuracy of Multi-Sea Winter (MSW) salmon is only briefly discussed. Expand on these findings with additional data or analysis.

Reply: The manuscript states the difference in scale reading accuracy for 1SW and MSW salmon with reference to summary statistics from the GLM where 1SW/MSW is added as a factor (L256). Accuracy for MSW salmon cannot be further explored due to the relative low sample size, especially for salmon with sea-age 3-5 year. This issue is now addressed in L415-418.

10. Provide more detail about the statistical methods used. For example: Explain why generalized logistic regression was chosen and how it was validated. Justify the use of Pearson correlation analysis and discuss its limitations in this context.

Reply: Sentences addressing these issues are added in L218-219 and L238. We believe it goes beyond the scope of this manuscript to explain why a generalized logistic regression were used when the response variable follows a binomial distribution. The description of model validation is partly modified. We have now used the R-package DHARMa to evaluate model performance and added a reference to this package in the manuscript. However, given that it’s the model follow a binomial distribution with only one final significant covariate, which is a factor, there is not much validation needed. A boxplot of the residuals for the two groups 1SW and MSW looks good and do not raise any concerns.

The reference to Pearson correlation given in the legend for figure 2 is removed in the revised manuscript.

11. The Discussion section mainly restates results without critically analyzing their implications. Include discussions on potential biases and limitations, such as reader variability or environmental factors affecting scale formation.

Reply: We agree. Sentence addressing these issues are added in L380-388.

12. Situate the study within the broader context of fisheries science. Compare the findings with previous research on Atlantic salmon or other salmonids, highlighting similarities and differences.

Reply: We have partly modified the text according to this reviewer comment. See L399-403. However, in our opinion, the discussion already discussed the findings in relation to the (few) other comparable studies. We have nevertheless added some more information and tried to better highlight the lack of other comparable study (L321-324 and 330-334).

---

## [Decision Letter · Decision Letter 1]

2 Apr 2025

Validating Atlantic salmon (Salmo Salar) scale reading by genetic parent assignment and PIT-tagging

PONE-D-24-56358R1

Dear Dr. Utne,

We’re pleased to inform you that your manuscript has been judged scientifically suitable for publication and will be formally accepted for publication once it meets all outstanding technical requirements.

Kind regards,

Tzong-Yueh Chen, Ph.D.

Academic Editor

PLOS ONE

Additional Editor Comments (optional):

Reviewers' comments:

Reviewer's Responses to Questions

**Comments to the Author**

1. If the authors have adequately addressed your comments raised in a previous round of review and you feel that this manuscript is now acceptable for publication, you may indicate that here to bypass the “Comments to the Author” section, enter your conflict of interest statement in the “Confidential to Editor” section, and submit your "Accept" recommendation.

Reviewer #1: All comments have been addressed

Reviewer #2: All comments have been addressed

2. Is the manuscript technically sound, and do the data support the conclusions?

Reviewer #1: Yes

Reviewer #2: Yes

3. Has the statistical analysis been performed appropriately and rigorously? 

Reviewer #1: Yes

Reviewer #2: Yes

4. Have the authors made all data underlying the findings in their manuscript fully available?

Reviewer #1: Yes

Reviewer #2: Yes

5. Is the manuscript presented in an intelligible fashion and written in standard English?

Reviewer #1: Yes

Reviewer #2: Yes

6. Review Comments to the Author

Reviewer #1: The authors have adequately addressed all my comments, and I find this manuscript acceptable for publication.

Reviewer #2: Overall, the authors have carefully addressed all review comments and have substantially improved the manuscript through appropriate revisions and clarifications. In particular, the authors have enhanced the clarity of the research objectives and provided additional details on sample characteristics, methodological procedures, and statistical analyses. The inclusion of new tables and figures has improved the transparency of the data and made the findings more accessible to readers.

The clarification of the scale reading calibration process and the discussion of limitations, such as the restricted availability of otoliths and the challenges with MSW salmon sample size, reflect a responsible and scientifically sound approach. While some suggestions—such as implementing alternative back-calculation methods—were acknowledged but not fully incorporated due to data constraints, the authors have provided reasonable justifications and expressed a willingness to explore these aspects in future studies.

In my opinion, the revised manuscript has demonstrated improved scientific rigor and greater applicability to fisheries science and management. The dataset and analytical approach are valuable contributions to the field, particularly for validating fish age determination methods using integrated genetic and tagging techniques. I believe the manuscript is now suitable for publication, pending the editor’s final decision.

7. PLOS authors have the option to publish the peer review history of their article (what does this mean? ). If published, this will include your full peer review and any attached files.

**Do you want your identity to be public for this peer review?** For information about this choice, including consent withdrawal, please see our Privacy Policy .

Reviewer #1: No

Reviewer #2: No

---

## [Editor Report · Acceptance letter]

PONE-D-24-56358R1

PLOS ONE

Dear Dr. Utne,

I'm pleased to inform you that your manuscript has been deemed suitable for publication in PLOS ONE. Congratulations! Your manuscript is now being handed over to our production team.

Kind regards,

on behalf of

Prof. Tzong-Yueh Chen

Academic Editor

PLOS ONE